# Multi-Robot Collision Avoidance under Uncertainty with Probabilistic Safety Barrier Certificates

**Wenhao Luo**[*]
The Robotics Institute
Carnegie Mellon University
wenhaol@cs.cmu.edu

**Wen Sun**
Computer Science Department
Cornell University
ws455@cornell.edu

**Ashish Kapoor**
Microsoft Corporation
Redmond, Washington 98033
akapoor@microsoft.com

## Abstract

Safety in terms of collision avoidance for multi-robot systems is a difficult challenge under uncertainty, non-determinism and lack of complete information. This paper aims to propose a collision avoidance method that accounts for both measurement uncertainty and motion uncertainty. In particular, we propose Probabilistic Safety Barrier Certificates (PrSBC) using Control Barrier Functions to define the space of admissible control actions that are probabilistically safe with formally provable theoretical guarantee. By formulating the chance constrained safety set into deterministic control constraints with PrSBC, the method entails minimally modifying an existing controller to determine an alternative safe controller via quadratic programming constrained to PrSBC constraints. The key advantage of the approach is that no assumptions about the form of uncertainty are required other than finite support, also enabling worst-case guarantees. We demonstrate effectiveness of the approach through experiments on realistic simulation environments.

## 1 Introduction

Safe control is one of the most important task that needs to be addressed in the realm of large-scale multi-robot systems. For example, consider the problem of building an automatic collision avoidance system (ACAS) for aerial robots that would scale up as the autonomous aerial traffic increases. Such a system needs to be computationally efficient for execution in real-time and robust to various real-world factors that include uncertainty, non-determinism and approximations made in the formulation of the system. Measurement uncertainty in the system arises from various estimation or prediction procedures in real-world that rely on sensory information ( e.g. LIDARS, on-board GPS) being collected in real-time to get robots state information. On the other hand, non-determinism often arises from our in-ability to model various exogenous variables that are part of our operating environment, e.g. phenomena such as wind gusts. Ability to pro-actively deal with such measurement and motion uncertainty is fundamental in the safety considerations.

In this work, we consider the problem of real-time safe control in terms of reactive collision avoidance for crowded multi-robot team operating in a realistic environment with their existing task-related controllers or control policies. Akin to real-world we consider scenarios with both measurement

---

[*]Work done while interning at Microsoft Corporation, Redmond.
  Code is available online at https://github.com/wenhaol/PrSBC.

uncertainty (e.g. noisy sensing and localization) and motion uncertainty (e.g. disturbances from environments and inaccurately modelled dynamics). Many safe control methods that attempt to address the measurement uncertainty often make restrictive assumptions, such as Gaussian representation of the uncertainties [9, 19, 26, 34, 35]. Approaches that consider bounded localization or control disturbance using conservative bounding volumes [8, 10, 13, 17] often overestimate the probability of collisions.

This paper proposes a novel approach that provides chance-constrained collision-free guarantees for multi-robot system under measurement and motion uncertainty. At the heart of the method is the idea of probabilistic safety barrier certificates (PrSBC) that enforces the chance constrained collision avoidance with deterministic constraints over an existing controller. With PrSBC constraints, the safety controller can be achieved by minimally modifying the existing controllers in real-time as done by other control barrier function approaches [4, 28]. This hence formally satisfies the collision-avoidance chance-constraints while staying as close to the original robot behaviors as possible. Our work is most closely related to the work on safety barrier certificates (SBC) for multi-robot collision avoidance [28] using permissive control barrier functions (CBF) [3, 4]. While the prior work focused on deterministic settings, our goal here is to provide a safety envelope around an existing controller that accounts for uncertainties and non-determinism in a probabilistic setting. There are several **advantages** of the proposed PrSBC. First, in contrast of other probabilistic collision avoidance approaches that directly constrain the inter-robot distance [30, 34, 35], the proposed method produces a more permissive set for the controllers with a tighter bound. Second, the PrSBC naturally inherits the forward invariance from CBF, e.g. robots staying in the collision-free set at all time, and thus enabling us to prove guarantees throughout the continuous time scale. Finally, it is natural to apply the chance constrained collision avoidance with PrSBC under both centralized and decentralized settings to bridge learning based methodologies and model based safety-critical control with provable safety guarantee. For example, one may use learning techniques such as Gaussian Processes to learn one or more partially unknown dynamical systems with noisy uncertainties and use our PrSBC approach to compute certified probablistically safe policies to collect more data for further improving models. We believe integrating dynamical system learning with our PrSBC framework to guarantee safe learning to control is an important future direction.

The key underlying assumption in our method is that the uncertainties arising due to sensor measurements, incomplete dynamics and other exogenous variables have finite support. This is a reasonable assumption for many of the multi-robot scenarios. For example, we can safely assume that true positions of robots, or the amount of wind gusts etc. are bounded within certain sensor specifications or physical parameters respectively (e.g. [11]). We use the task similar to automatic collision avoidance system for aerial robots as a motivating application. Our experiments explore the proposed computation of PrSBC controller in both centralized and decentralized settings, which can handle both the uncertainties as well as environmental disturbances while continuously guaranteeing safety. In summary, the **core contributions of this paper** are as follows:

1. A novel chance-constrained collision avoidance method with Probabilistic Safety Barrier Certificates (PrSBC) ensuring provable forward invariance under uncertainties with bounded support.

2. Formal proof of existence of PrSBC in a closed form.

3. Experimental results on the task similar to automatic collision avoidance for aerial robots that demonstrate efficiency, scalability and distributed computation.

## 2 Related Work

Collision avoidance for multi-robot system operating in dynamic environments have been studied for safety consideration over the years. Reactive methods such as reciprocal velocity obstacles (RVO) [1, 2, 23, 25], safety barrier certificates (SBC) [6, 27, 28], and buffered Voronoi cells [33] have been presented to compute on-line multi-robot collision-free motions in a distributed manner. To account for uncertainties associated to the robot state information and motion model, the deterministic collision avoidance methods have been extended to probabilistic representations [9, 12] with chance constraints formulation [30, 34, 35] assuming Gaussian representation of uncertainties in order to derive close form solution. The concept of velocity obstacles is adopted to develop enlarged conservative bounding volumes around the robot [8, 10, 13, 17]. It remains challenging when prior

knowledge of the uncertainty model is not available or it is not necessarily Gaussian, e.g. readings from an on-board GPS sensor that only have an expected value with an finite support as accuracy.

Another family of reactive collision avoidance approaches is the recent optimization-based safety control using control barrier function [3, 4, 16, 27, 28, 31]. The safety controller is able to minimally revise the nominal controller in the context of quadratic programming and ensures the robots remain in the safety set at all time, leading to a minimally invasive safe control behavior. In [28], the control barrier function is employed to develop the Safety Barrier Certificates (SBC) for multi-robot systems, depicting a non-conservative safety envelope for the multi-robot controller from which the robots stay collision-free at all time. Extensions to higher order nonlinear system dynamics using SBC and Exponential Control Barrier Function (ECBF) have been introduced in [16, 27, 29]. Very recently, safe learning using CBF and ECBF for safety consideration are presented in [14, 20, 22, 29] that learns the motion disturbance or partially known dynamics with *perfect* localization information. In this paper, we propose the probabilistic safety barrier certificates (PrSBC) for multi-robot systems which extends the deterministic SBC [28] to a probabilistic setting to account for both localization and motion uncertainties of the ego robot and other robots/obstacles. No assumptions about the uncertainty model are required other than finite support. We discuss in Section 5 that the PrSBC could handle other uncertainty models as well, and hence is useful for uncertainty-aware safe learning applications in the future work.

## 3  Problem Statement

### 3.1  Robot and obstacle model

Consider a team of $N$ robots moving in a shared $d$-dimensional workspace. Each robot $i \in \mathcal{I} = \{1, \ldots, N\}$ is centered at the position $\mathbf{x}_i \in \mathcal{X}_i \subset \mathbb{R}^d$ and enclosed with a uniform safety radius $R_i \in \mathbb{R}$. The stochastic dynamical system $\dot{\mathbf{x}}_i$ in control affine form with noise and the noisy observation $\hat{\mathbf{x}}_i \in \mathbb{R}^d$ of each robot $i$ are described as follows.

$$\begin{aligned} \dot{\mathbf{x}}_i &= f_i(\mathbf{x}_i, \mathbf{u}_i) + \mathbf{w}_i = F_i(\mathbf{x}_i) + G_i(\mathbf{x}_i)\mathbf{u}_i + \mathbf{w}_i , \quad \mathbf{w}_i \sim U(-\Delta\mathbf{w}_i, \Delta\mathbf{w}_i) \\ \hat{\mathbf{x}}_i &= \mathbf{x}_i + \mathbf{v}_i , \quad \mathbf{v}_i \sim U(-\Delta\mathbf{v}_i, \Delta\mathbf{v}_i) \end{aligned} \quad (1)$$

where $\mathbf{u}_i \in \mathcal{U}_i \subseteq \mathbb{R}^m$ denotes the control input. $F_i$ and $G_i$ are locally Lipschitz continuous. The deterministic system dynamics $f_i(\mathbf{x}_i, \mathbf{u}_i) = F_i(\mathbf{x}_i) + G_i(\mathbf{x}_i)\mathbf{u}_i$ in control affine form is general and could describe a large family of nonlinear systems, e.g. 3-dof differential drive vehicles with unicycle dynamics ([18, 28]), 12-dof quadrotors with underactuated system ([29, 32]), bipedal robots, automotive vehicle, and Segway robots [4, 22]. $\mathbf{w}_i, \mathbf{v}_i \in \mathbb{R}^d$ are the uniformly distributed process noise and the measurement noise respectively and considered as continuous independent random variables with finite support. A uniform distribution is a natural choice for these noise processes, however, most of our analysis does not require the exact form except that the support is finite. This finite support can vary at each time-point and come from a perception module, a state estimator or other physical parameters of the system.

**Obstacle Model:** Similar to the robots, other static or moving obstacles $k \in \mathcal{O} = \{1, \ldots, K\}$ are also modeled as a rigid sphere located at $\mathbf{x}_k \in \mathbb{R}^d$ with the safety radius $R_k \in \mathbb{R}$. The measurement of obstacle location via robot sensor is modeled as $\hat{\mathbf{x}}_k = \mathbf{x}_k + \mathbf{v}_k \in \mathbb{R}^d$ with bounded uniformly distributed noise $\mathbf{v}_k \sim U(-\Delta\mathbf{v}_k, \Delta\mathbf{v}_k)$. As commonly assumed in other collision avoidance work ([9, 20, 35]), we consider the piece-wise constant obstacle' velocity to be detected by the robots as $\hat{\mathbf{u}}_k$ with a bounded noise, rendering the obstacle dynamics as $\dot{\mathbf{x}}_k = \mathbf{u}_k = \hat{\mathbf{u}}_k + \mathbf{w}_k \in \mathbb{R}^d$, $\mathbf{w}_k \sim U(-\Delta\mathbf{w}_k, \Delta\mathbf{w}_k)$. The finite supports of $\mathbf{v}_k, \mathbf{w}_k$ are also assumed to be known by the robots.

Denote the joint robot states as $\mathbf{x} = \{\mathbf{x}_1, \ldots, \mathbf{x}_N\} \in \mathcal{X} \subset \mathbb{R}^{d \times N}$ and the joint obstacle states as $\mathbf{x}_o = \{\mathbf{x}_1, \ldots, \mathbf{x}_K\} \in \mathcal{X}_o \in \mathbb{R}^{d \times K}$. For any pair-wise inter-robot or robot-obstacle collision avoidance between robots $i, j \in \mathcal{I}$ and obstacles $k \in \mathcal{O}$, the safety of $\mathbf{x}$ is defined as follows.

$$h_{i,j}^s(\mathbf{x}) = \|\mathbf{x}_i - \mathbf{x}_j\|^2 - (R_i + R_j)^2 , \ \forall i > j , \quad h_{i,k}^s(\mathbf{x}, \mathbf{x}_o) = \|\mathbf{x}_i - \mathbf{x}_k\|^2 - (R_i + R_k)^2 , \ \forall i, k \quad (2)$$

$$\mathcal{H}_{i,j}^s = \{\mathbf{x} \in \mathbb{R}^{d \times N} : h_{i,j}^s(\mathbf{x}) \geq 0\} , \ \forall i > j , \quad \mathcal{H}_{i,k}^s = \{\mathbf{x} \in \mathbb{R}^{d \times N} : h_{i,k}^s(\mathbf{x}, \mathbf{x}_o) \geq 0\} , \ \forall i, k \quad (3)$$

### 3.2  Chance-constrained collision avoidance for safety

As the robots only have access to the noisy measurements on the states of the robots and obstacles, the positions of the robots and obstacles are modeled as random variables with a finite support.

The collision avoidance constraints can then be considered in a chance-constrained setting for each pairwise robots $i, j$ and robot-obstacle $i, k$. Formally, given the minimum admissible probability of safety $\sigma, \sigma_o \in [0, 1]$ predefined by the user, it is required that:

$$\Pr(\mathbf{x}_i, \mathbf{x}_j \in \mathcal{H}_{i,j}^s) \geq \sigma, \quad \forall i > j, \qquad \Pr(\mathbf{x}_i, \mathbf{x}_k \in \mathcal{H}_{i,k}^s) \geq \sigma_o, \quad \forall i, k \qquad (4)$$

$\Pr(\cdot)$ indicates the probability of an event. Note that when $\sigma, \sigma_o$ are set to 1, the conditions naturally lead to the worst-case collision avoidance with enlarged bounded volume as discussed in section 5. Such worst-case guarantees can lead to a conservative behavior, thus often there are advantages in maintaining a probabilistic safety.

Assume that each robot has a task-related controller $\mathbf{u}_i^* \in \mathbb{R}^m$. We consider the chance-constrained collision avoidance as a one-step optimization problem that minimally modifies $\mathbf{u}_i^*$ for each robot $i$, while satisfying the desired probabilistic safety in (4). Formally we solve the following Quadratic Program (QP) under the safety constraints:

$$\min_{\mathbf{u} \in \mathbb{R}^{mN}} \sum_{i=1}^{N} \|\mathbf{u}_i - \mathbf{u}_i^*\|^2 \quad \text{subject to (4) and } \|\mathbf{u}_i\| \leq \alpha_i, \forall i \in \{1, \dots, N\} \qquad (5)$$

where $\mathbf{u} \in \mathcal{U} \subset \mathbb{R}^{mN}$ is the bounded joint control input of all the robots with bounded magnitude $\alpha_i, \forall i$. Next, we briefly describe the background of Safety Barrier Certificates (SBC) [28]. Section 5 then presents our method of Probabilistic Safety Barrier Certificates (PrSBC) that utilizes control barrier functions [4] to remap the probabilistic safety set constraints (4) from the state space $\mathcal{X} \subset \mathbb{R}^{d \times N}$ to the control space $\mathcal{U} \subset \mathbb{R}^{mN}$.

## 4   Background: Safety Barrier Certificates (SBC)

In this section we review the formulation of the *deterministic* safety control constraints. Without loss of generality, we represent a desired safety set $\mathcal{H}^s$ using function $h^s(\mathbf{x})$ as:

$$\mathcal{H}^s = \{\mathbf{x} \in \mathbb{R}^{d \times N} \mid h^s(\mathbf{x}) \geq 0\} \qquad (6)$$

We summarize the conditions on controllers $\mathbf{u} \in \mathcal{U} \subseteq \mathbb{R}^{mN}$ based on Zeroing Control Barrier Functions (ZCBF) ([3]) and the Safety Barrier Certificates (SBC) ([28]) to guarantee *forward invariance* of safety. Formally, a safety condition is forward-invariant if $\mathbf{x}(t = 0) \in \mathcal{H}^s$ implies $\mathbf{x}(t) \in \mathcal{H}^s$ for all $t > 0$ with the designed satisfying controller at each time step. The Theorem of ZCBF and forward invariance from [3, 28] is summarized as the following Lemma.

**Lemma 1.** *Given the dynamical system in equ. (1) without uncertainties, i.e. $\mathbf{w}_i = 0, \forall i \in \mathcal{I}$ and the set $\mathcal{H}^s$ defined by equ. (6) for the continuously differentiable function $h^s : \mathbb{R}^{d \times N} \to \mathbb{R}$. The function $h^s$ is a ZCBF and the admissible control space $S(\mathbf{x})$ for each time step can be defined as*

$$S(\mathbf{x}) = \{\mathbf{u} \in \mathcal{U} \mid \dot{h}^s(\mathbf{x}, \mathbf{u}) + \kappa(h^s(\mathbf{x})) \geq 0\}, \ \mathbf{x} \in \mathcal{X}, \qquad (7)$$

*where $\kappa$ is an extended class-$\mathcal{K}$ function. Then any Lipschitz continuous controller satisfying $\mathbf{u} \in S(\mathbf{x})$ at each time step for the system (1) renders the set $\mathcal{H}^s$ forward invariant, i.e. robots stay collision-free at all times.*

As described in ([28]), the extended class-$\mathcal{K}$ function $\kappa$ such as $\kappa(r) = r^P$ with any positive odd integer $P$ leads to different behaviors of the state of the system approaching the boundary of safety set $\mathcal{H}^s$ in (6). Similar to ([28]), in this paper we use the particular choice of $\kappa(h^s(\mathbf{x})) = \gamma h^s(\mathbf{x})$ with $\gamma > 0$. In order to render a larger admissible control space $S(x)$, a very large value of $\gamma >> 0$ will be adopted. Thus the admissible control space in (7) induces the following pairwise constraints over the controllers, referred as SBC ([28]):

$$\mathcal{B}^s(\mathbf{x}) = \{\mathbf{u} \in \mathbb{R}^{mN} : \dot{h}_{i,j}^s(\mathbf{x}, \mathbf{u}) + \gamma h_{i,j}^s(\mathbf{x}) \geq 0, \ \forall i > j\}$$
$$\mathcal{B}^o(\mathbf{x}, \mathbf{x}_o) = \{\mathbf{u} \in \mathbb{R}^{mN} : \dot{h}_{i,k}^s(\mathbf{x}, \mathbf{x}_o, \mathbf{u}, \mathbf{u}^o) + \gamma h_{i,k}^s(\mathbf{x}, \mathbf{x}_o) \geq 0, \ \forall i, k\} \qquad (8)$$

where $\mathbf{u}^o \in \mathbb{R}^{dK}$ is the joint control input of all the obstacles not controllable by the robots. Here $\mathcal{B}^s(\mathbf{x}), \mathcal{B}^o(\mathbf{x}, \mathbf{x}_o)$ define the SBC for the inter-robot and robot-obstacle collision avoidance respectively, rendering the safety set $\mathcal{H}^s$ forward invariant: the robots will always stay safe, i.e. satisfying (3) at all times if they are initially collision free and the robots' joint control input $\mathbf{u}$ lies in the set $\mathcal{B}^s(\mathbf{x}) \cap \mathcal{B}^o(\mathbf{x}, \mathbf{x}_o)$. One of the useful properties of (8) is that they induce linear constraints over both the pair-wise control inputs $\mathbf{u}_i$ and $\mathbf{u}_j$ (inter-robot) and control input $\mathbf{u}_i$ (robot-obstacle).

# 5 Probabilistic Safety Barrier Certificates

## 5.1 Probabilistic Safety Barrier Certificates (PrSBC)

We seek a probabilistic version of Lemma 1 that implies the SBC in (8) as a sufficient condition for the forward invariance of $\mathcal{H}^s$ in (6). Given the assumption that each pairwise robots are initially collision-free, i.e. $\mathbf{x}_i, \mathbf{x}_j \in \mathcal{H}^s_{i,j}$ at $t = 0$ and the sufficiency condition in Lemma 1, we have $\mathbf{u}_i, \mathbf{u}_j \in \mathcal{B}^s_{i,j}(\mathbf{x}) \implies \mathbf{x}_i, \mathbf{x}_j \in \mathcal{H}^s_{i,j}$ and $\mathbf{u}_i, \mathbf{u}_j \notin \mathcal{B}^s_{i,j}(\mathbf{x}) \implies \mathbf{x}_i, \mathbf{x}_j \notin \mathcal{H}^s_{i,j}$. Hence it is straightforward to show that $\Pr(\mathbf{u}_i, \mathbf{u}_j \in \mathcal{B}^s_{i,j}(\mathbf{x})) \leq \Pr(\mathbf{x}_i, \mathbf{x}_j \in \mathcal{H}^s_{i,j})$ and $\Pr(\mathbf{u}_i, \mathbf{u}_k \in \mathcal{B}^o_{i,k}(\mathbf{x}, \mathbf{x}_o)) \leq \Pr(\mathbf{x}_i, \mathbf{x}_k \in \mathcal{H}^s_{i,k})$. Consequently, we can derive the following inter-robot and robot-obstacle probabilistic collision free sufficiency conditions corresponding to equ. (4):

$$\begin{aligned}
\Pr(\mathbf{u}_i, \mathbf{u}_j \in \mathcal{B}^s_{i,j}(\mathbf{x})) \geq \sigma &\implies \Pr(\mathbf{x}_i, \mathbf{x}_j \in \mathcal{H}^s_{i,j}) \geq \sigma, \quad \forall i > j \\
\Pr(\mathbf{u}_i, \mathbf{u}_k \in \mathcal{B}^o_{i,k}(\mathbf{x}, \mathbf{x}_o)) \geq \sigma_o &\implies \Pr(\mathbf{x}_i, \mathbf{x}_k \in \mathcal{H}^s_{i,k}) \geq \sigma_o, \quad \forall i, \ k
\end{aligned} \tag{9}$$

Given these reformulated collision-free chance constraints over controllers, we now formally define the Probabilistic Safety Barrier Certificates (PrSBC):

**Definition 2.** *Probabilistic Safety Barrier Certificates (PrSBC): Given a confidence level $\sigma \in [0, 1]$, PrSBC determines the admissible control space $\mathcal{S}^\sigma_u$ at each time-step guaranteeing the chance-constrained safety condition in equ. (4) and are defined as the intersection of $n$ different half-spaces where $n$ is the total number of pairwise deterministic inter-robot constraints.*

$$\mathcal{S}^\sigma_u = \{\mathbf{u} \in \mathbb{R}^{mN} \mid A^\sigma_{ij}\mathbf{u} \leq b^\sigma_{ij}, \quad \forall i > j, \ A^\sigma \in \mathbb{R}^{n \times mN}, \ b^\sigma \in \mathbb{R}^n\} \tag{10}$$

Here we first introduce the definition and form of PrSBC. The computation of $A^\sigma \in \mathbb{R}^{n \times mN}, b^\sigma \in \mathbb{R}^n$ determined by $\sigma$ will be given in the latter part of equ. (13) and (14) for inter-robot and robot-obstacle collision avoidance. The PrSBC hence characterizes the admissible safe control space for the multi-robot team with probabilistic safety guarantee.

**Theorem 3.** *Existence of PrSBC: Assuming all pairwise robots are initially collision-free at $t = 0$, i.e. equ. (2) holds true for all possible value of random state variables $\mathbf{x}_i \in [\hat{\mathbf{x}}_i - \Delta\mathbf{v}_i, \hat{\mathbf{x}}_i + \Delta\mathbf{v}_i], \forall i \in \mathcal{I}$, then the PrSBC defined in equ. (10) is guaranteed to exist for any given confidence level $\sigma \in [0, 1]$.*

*Proof.* The detailed proof is provided in the Appendix. The basic idea is to start by proving the existence of PrSBC between each pairwise robots $i$ and $j$ under any user-defined confidence level $\sigma \in [0, 1]$. Consider $\dot{h}^s_{i,j}(\mathbf{x}, \mathbf{u}) = \frac{\partial h^s_{i,j}}{\partial \mathbf{x}}(\mathbf{x})(\Delta F_{i,j}(\mathbf{x}) + G_{i,j}(\mathbf{x})\mathbf{u}_{i,j} + \Delta\mathbf{w}_{i,j})$ with $\Delta F_{i,j}(\mathbf{x}) = F_i(\mathbf{x}_i) - F_j(\mathbf{x}_j)$, $G_{i,j}(\mathbf{x})\mathbf{u}_{i,j} = G_i(\mathbf{x}_i)\mathbf{u}_i - G_j(\mathbf{x}_j)\mathbf{u}_j$, and $\Delta\mathbf{w}_{i,j} = \mathbf{w}_i - \mathbf{w}_j$. We can re-write the sufficiency condition $\Pr(\mathbf{u}_i, \mathbf{u}_j \in \mathcal{B}^s_{i,j}(\mathbf{x})) \geq \sigma$ in (9) using (8) as follows:

$$\Pr(\mathbf{u}_i, \mathbf{u}_j \in \mathcal{B}^s_{i,j}(\mathbf{x})) \geq \sigma :$$
$$\iff \Pr\left(\frac{\partial h^s_{i,j}}{\partial \mathbf{x}}(\mathbf{x})G_{i,j}(\mathbf{x})\mathbf{u}_{i,j} \geq -\gamma h^s_{i,j}(\mathbf{x}) - \frac{\partial h^s_{i,j}}{\partial \mathbf{x}}(\mathbf{x})\big(\Delta F_{i,j}(\mathbf{x}) + \Delta\mathbf{w}_{i,j}\big)\right) \geq \sigma \tag{11}$$

This is a chance constraint over pairwise controller $\mathbf{u}_i, \mathbf{u}_j$. Note that $\mathbf{x} = \hat{\mathbf{x}} - \mathbf{v} \in \mathbb{R}^{d \times N}$ is random variable with finite support and $\hat{\mathbf{x}}, \mathbf{v} \in \mathbb{R}^{d \times N}$ are the joint observation and joint measurement noise respectively. Since it is assumed all pairwise robots are initially collision-free at current time step $t = 0$ and the robot locations are noisy but bounded with finite support, via Bayesian decomposition we are able to prove there always exists a solution of pairwise $\mathbf{u}_i, \mathbf{u}_j$ rendering $\Pr(\mathbf{u}_i, \mathbf{u}_j \in \mathcal{B}^s_{i,j}(\mathbf{x})) = \sigma$ in (11) and the non-empty admissible control space about $\mathbf{u}_i, \mathbf{u}_j$ for $\Pr(\mathbf{u}_i, \mathbf{u}_j \in \mathcal{B}^s_{i,j}(\mathbf{x})) \geq \sigma$. It is then straightforward to extend to all pairwise inter-robot collision avoidance constraints and thus concludes the proof. $\square$

**Computation of PrSBC:** Lets consider inter-robot collision avoidance first. Given any confidence level $\sigma \in [0, 1]$, the equivalent chance constraint of $\Pr(\mathbf{u}_i, \mathbf{u}_j \in \mathcal{B}^s_{i,j}(\mathbf{x})) \geq \sigma$ in (11) can be transformed into a deterministic linear constraint over pairwise controllers $\mathbf{u}_i, \mathbf{u}_j$ in the form of (10). We obtain an approximate solution by considering the condition on each individual dimension $\forall l = 1, \ldots, d$. To simplify the discussion, we assume $\sigma > 0.5$ and denote $e^{l,1}_{i,j} = \Phi^{-1}(\sigma)$ and $e^{l,2}_{i,j} = \Phi^{-1}(1 - \sigma)$ with $\Phi^{-1}(\cdot)$ as the inverse cumulative distribution function (CDF) of the random

variable $\Delta \mathbf{x}_{i,j}^l = \mathbf{x}_i^l - \mathbf{x}_j^l$ in the $l$th dimension. We have $\sigma > 0.5 \implies e_{i,j}^{l,1} > e_{i,j}^{l,2}$. Thus, the sufficient condition for (11) formally becomes the following deterministic constraint (detailed deduction is provided in the Appendix):

$$\exists l = 1, \ldots, d: \quad -2e_{i,j}^l (G_i \mathbf{u}_i - G_j \mathbf{u}_j)_l / \gamma \le (e_{i,j}^l)^2 - R_{ij}^2 + B_{ij}^l + 2e_{i,j}^l \Delta F_{i,j}^l / \gamma \quad (12)$$

where $R_{ij} = R_i + R_j$ and $B_{ij}^l$ is a constant determined by finite support of $\mathbf{w}_i, \mathbf{w}_j, \Delta \mathbf{x}_{i,j}$ at the $l$th dimension. To simplify the discussion we assume piece-wise $G_i, G_j \in \mathbb{R}^{d \times m}, F_i, F_j \in \mathbb{R}^{d \times 1}$ in (1) are known and deterministic. $(G_i \mathbf{u}_i - G_j \mathbf{u}_j)_l, \Delta F_{i,j}^l \in \mathbb{R}$ denote the $l$th element of $(G_i \mathbf{u}_i - G_j \mathbf{u}_j) \in \mathbb{R}^{d \times 1}$ and $\Delta F_{i,j} = F_i - F_j \in \mathbb{R}^{d \times 1}$ respectively. Also, we have $e_{i,j}^l = e_{i,j}^{l,2}$ if $e_{i,j}^{l,2} > 0$, or $e_{i,j}^{l,1}$ if $e_{i,j}^{l,1} < 0$, or 0 if $e_{i,j}^{l,2} \le 0$ and $e_{i,j}^{l,1} \ge 0$. Note that $e_{i,j}^l = 0$ implies the two robots $i$ and $j$ overlap along the $l$th dimension, e.g. two drones flying to the same 2D locations but with different altitudes. As it is assumed any pairwise robots are initially collision free and from the forward invariance property discussed above, $e_{i,j}^l = 0$ only happens along at most $d-1$ dimensions. To that end, we can formally construct the PrSBC in (10) with the following linear deterministic constraints in closed form.

$$\mathcal{S}_u^\sigma = \{\mathbf{u} \in \mathbb{R}^{mN} | -2\mathbf{e}_{i,j}^T (G_i \mathbf{u}_i - G_j \mathbf{u}_j)/\gamma \le ||\mathbf{e}_{i,j}||^2 - d \cdot R_{ij}^2 + B_{ij} + 2\mathbf{e}_{i,j}^T \Delta F_{i,j}/\gamma, \quad \forall i > j\} \quad (13)$$

where $\mathbf{e}_{i,j} = [\mathbf{e}_{i,j}^1, \ldots, \mathbf{e}_{i,j}^d]^T \in \mathbb{R}^{d \times 1}$ and $B_{ij} = \Sigma_{l=1}^d B_{ij}^l$. This invokes a set of pairwise linear constraints over the robot controllers such that the inter-robot probabilistic collision avoidance in (4) holds true at all times. Note the PrSBC constraint in (13) is a conservative approximation of (12) and therefore guarantee $\Pr(\mathbf{u}_i, \mathbf{u}_j \in \mathcal{B}_{i,j}^s(\mathbf{x})) \ge \sigma$. Please see the detailed discussion in the Appendix.

**Remark 1.** *For other forms of distribution than uniform but with finite support for the noise models, the only change is the computation of inverse CDF to specify different $e_{i,j}^{l,1}, e_{i,j}^{l,2}$ and the rest of the derivations of PrSBC still holds and ensure chance-constrained safety. For Gaussian distribution with infinite support, we can still compute a finite support based on the corresponding inverse CDF from $\sigma$ for $\Delta \mathbf{x}_{i,j}$ at each dimension.*

**Proposition 4.** ***PrSBC for Robot-Obstacle Collision Avoidance:*** *Consider the dynamic obstacle model described in Section. 3.1 and PrSBC for pairwise robots in (13), the PrSBC for robot-obstacle collision avoidance with a given confidence level $\sigma_o \in [0, 1]$ can be defined as follows.*

$$\mathcal{S}_u^{\sigma_o} = \{\mathbf{u} \in \mathbb{R}^{mN} | -2\mathbf{e}_{i,k}'^T G_i \mathbf{u}_i / \gamma \le -2\mathbf{e}_{i,k}'^T \hat{\mathbf{u}}_k / \gamma + ||\mathbf{e}_{i,k}'||^2 - d \cdot R_{ik}^2 + B_{ik} + 2\mathbf{e}_{i,k}'^T F_i / \gamma, \forall i, k\} \quad (14)$$

*where the intermediate variables of $\mathbf{e}_{i,k}', B_{ik}$ are computed the same way as for inter-robot case (13).*

**Proposition 5.** *PrSBC in (13) can be considered as a generalized SBC when the dynamics model in (1) is deterministic and without any uncertainty, i.e. $\mathbf{w}, \mathbf{v} = 0$. In this case, we have $\mathbf{e}_{i,j} = \Delta \mathbf{x}_{i,j} = \Delta \hat{\mathbf{x}}_{i,j} = \hat{\mathbf{x}}_i - \hat{\mathbf{x}}_j$ and $B_{ij} = 0$ in (13), and then it degenerates to the constraint in (8) same as SBC ([28]).*

**Proposition 6.** ***Worst-case Collision Avoidance:*** *when confidence level is set to be $\sigma = 1$, the PrSBC in (13) hence leads to the worst-case driven collision avoidance with $\mathbf{e}_{i,j}$ specified by the boundary of finite support of $\Delta \mathbf{x}_{i,j}$, yielding most conservative motions of $\mathbf{u}$ for all of the robots.*

## 5.2 Optimization-based Controllers with Probabilistic Safety Barrier Certificates

The constrained control space specified by PrSBC in (13) and (14) ensures the forward invariance of probabilistic safety in (9). Hence, we can reformulate the original QP problem in (5) with the PrSBC constraints as follows to obtain the probabilistic safety controller.

$$\mathbf{u} = \arg\min_{\mathbf{u} \in \mathbb{R}^{mN}} \sum_{i=1}^N ||\mathbf{u}_i - \mathbf{u}_i^*||^2 \quad \text{subject to } \mathbf{u} \in \mathcal{S}_\mathbf{u}^\sigma \bigcap \mathcal{S}_\mathbf{u}^{\sigma_o}, \quad ||\mathbf{u}_i|| \le \alpha_i, \forall i = 1, \ldots, N \quad (15)$$

Note that PrSBC constraints invoke a set of linear constraints over controllers and hence the probabilistic safety controller (15) can be solved in real-time with guaranteed probability of safety. The resulting safe controller per time step ensures for all $t \in [0, \tau]$, $\mathbf{u} \in \mathcal{S}_\mathbf{u}^\sigma \bigcap \mathcal{S}_\mathbf{u}^{\sigma_o}$, then our approach guarantees chance constrained safety along the entire time horizon $[0, \tau]$.

**Remark 2.** *(Probability of collision for the full trajectory) Denoting $n_t$ as the total number of time steps during execution, the probability of collision avoidance between robot $i, j$ for the whole*

*trajectory is lower bounded as $Pr\left(\bigcap_{t=1}^{n_t}(\mathbf{x}_i^t, \mathbf{x}_j^t \in \mathcal{H}_{i,j}^s(t))\right) = \prod_{t=1}^{n_t} Pr(\mathbf{x}_i^t, \mathbf{x}_j^t \in \mathcal{H}_{i,j}^s(t)) \geq \sigma^{n_t}$. Here we assume the probability of collision avoidance at each time step is independent for practical purposes as done in [24, 35]. In theory, by selecting $\sigma = exp(\frac{ln\sigma_{all}}{n_t})$ one could achieve a lower bounded joint collision free threshold of $\sigma_{all}$ for the full trajectory. However, it could be over-conservative in the long run, and hence we use step-wise threshold to construct local collision constraints. An alternative is to impose discounting factor $\beta < 1$ so that the penalty of future violation probabilities is relaxed, i.e. step-wise threshold $\sigma$ renders the same bounded joint threshold for the whole trajectory $\sum_{t=1}^{n_t}(\beta)^t Pr(\mathbf{x}_i^t, \mathbf{x}_j^t \in \mathcal{H}_{i,j}^s(t)) \geq \sigma$ if given discounting factor $\beta > 0.5$ (see [35]).*

### 5.3  Decentralized Probabilistic Safety Controller

While the controller (15) is centralized, we can also derive a decentralized version of the PrSBC and the controllers. The mechanism is similar to [28] which was originally applied to deterministic SBC.

Consider the PrSBC in equ. (13) and denote $b_{ij}^\sigma = ||\mathbf{e}_{i,j}||^2 - d \cdot R_{ij}^2 + B_{ij} + 2\mathbf{e}_{i,j}^T \Delta F_{i,j}/\gamma$. We can separate the linear pairwise PrSBC constraint between robot $i$ and $j$ in the following two inequalities:

$$-2\mathbf{e}_{i,j}^T G_i/\gamma \cdot \mathbf{u}_i \leq p_{ij}/(p_{ij} + p_{ji}) \cdot b_{ij}^\sigma, \quad 2\mathbf{e}_{i,j}^T G_j/\gamma \cdot \mathbf{u}_j \leq p_{ji}/(p_{ij} + p_{ji}) \cdot b_{ij}^\sigma. \quad (16)$$

Here $p_{ij}, p_{ji} \in [0, 1]$ represents the responsibility that each of the two robot takes regarding satisfying this pairwise probabilistic safety constraint. The knowledge of $p_{ij}, p_{ji}$ can be either predefined and assumed known by all robots, in which case each robot does not need to communicate and simply avoid collision in a reciprocal manner, or can be communicated locally between pairwise robots in a more cooperative manner. Note that equ. (16) is a sufficient condition of equ. (13) and hence still guarantees the required probabilistic safety.

With such decentralized constraints, we have the decentralized probabilistic safety controller:

$$\mathbf{u}_i = \underset{\mathbf{u}_i \in \mathbb{R}^m}{\arg\min} ||\mathbf{u}_i - \mathbf{u}_i^*||^2 \text{ subject to } \quad \mathbf{u}_i \in \mathcal{S}_{\mathbf{u}_i}^\sigma \bigcap \mathcal{S}_{\mathbf{u}_i}^{\sigma_o}, \quad ||\mathbf{u}_i|| \leq \alpha_i \quad (17)$$

with $\mathcal{S}_{\mathbf{u}_i}^\sigma = \{\mathbf{u}_i \in \mathbb{R}^m | -2\mathbf{e}_{i,j}^T G_i/\gamma \cdot \mathbf{u}_i \leq p_{ij}/(p_{ij} + p_{ji}) \cdot b_{ij}^\sigma, \forall j \in \mathcal{N}_i\}$ and $\mathcal{S}_{\mathbf{u}_i}^{\sigma_o} = \{\mathbf{u}_i \in \mathbb{R}^m | -2\mathbf{e}_{i,k}'^T G_i \mathbf{u}_i/\gamma \leq -2\mathbf{e}_{i,k}'^T \hat{\mathbf{u}}_k/\gamma + ||\mathbf{e}_{i,k}'||^2 - R_{ik}^2 + B_{ik} + 2\mathbf{e}_{i,k}'^T F_i/\gamma, \forall k \in \mathcal{K}\}$. $\mathcal{N}_i$ denotes the set of neighboring robots around robot $i$.

This decentralized PrSBC controller does not require centralized optimization process as for (15), but may thus lead to more conservative motion of robots or infeasible solution in extreme cases. In this case the robots will simply decelerate to zero velocities to ensure safety, which may cause the deadlock preventing the robots from achieving the goals. Some deconfliction policies for deterministic SBC can thereby be employed as in [7]. Readers are referred to [7] for detailed solutions.

## 6  Experimental Evaluation and Results

**Simulation Example:** Fig. 1 demonstrates the first set of simulations performed on a team of $N = 6$ mobile robots with unicycle dynamics using our PrSBC from (15) and the comparing deterministic SBC from [28], with both in centralized setting. We employ nonlinear inversion method ([18]) to map the desired velocity to the unicycle dynamics of mobile robots without compromising the safety guarantee. In this example, all of the robots use the gradient based controller $\mathbf{u}_i^* = -K_p(\mathbf{x}_i - \mathbf{x}_{i,goal})$ as the nominal control input to swap their positions with the robot on the opposite side, e.g. robot 1 with 2, 3 with 4, and 5 with 6 shown in Fig. 1a. Locations indexed in red are the goal positions for the corresponding robots. The robot safety radius is set to be $R_i = 0.2$m and has bounded uniformly distributed localization error denoted by the red error box accounting for the safety radius. At each time step, every robot only has access to the noisy measurement marked by dashed black circle covering each robot. Maximum velocity limit is 0.1m/sec for the robots and robots motion is disturbed by randomly generated bounded noise with magnitude up to 0.07m/sec. The inter-robot collision-free confidence level $\sigma$ set to be 0.9. Code is available online at https://github.com/wenhaol/PrSBC.

As SBC [28] is designed for a deterministic system, here it takes the noisy measurement of the robots directly as the robot states to compose the SBC for collision avoidance controller. We observe from Fig. 1f that collisions occur (robot 1 and 5) due to uncertainty in measured robot states as well as the

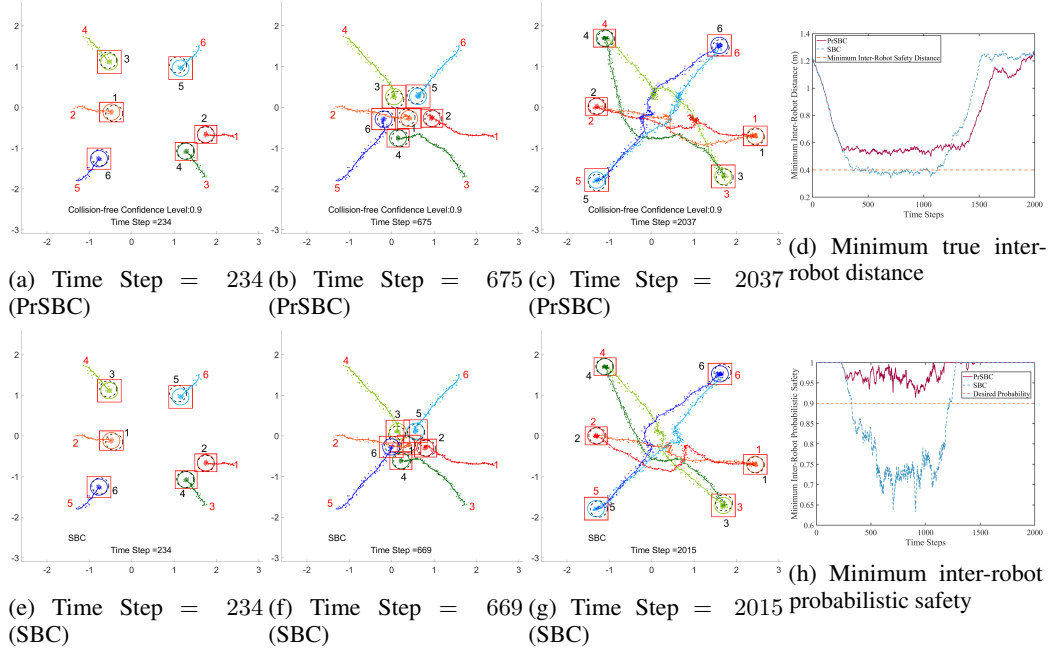

(a) Time Step = 234 (PrSBC)   (b) Time Step = 675 (PrSBC)   (c) Time Step = 2037 (PrSBC)   (d) Minimum true inter-robot distance

(e) Time Step = 234 (SBC)   (f) Time Step = 669 (SBC)   (g) Time Step = 2015 (SBC)   (h) Minimum inter-robot probabilistic safety

Figure 1: Simulation example of 6 robots swapping positions with collision-free confidence level $\sigma = 0.9$.

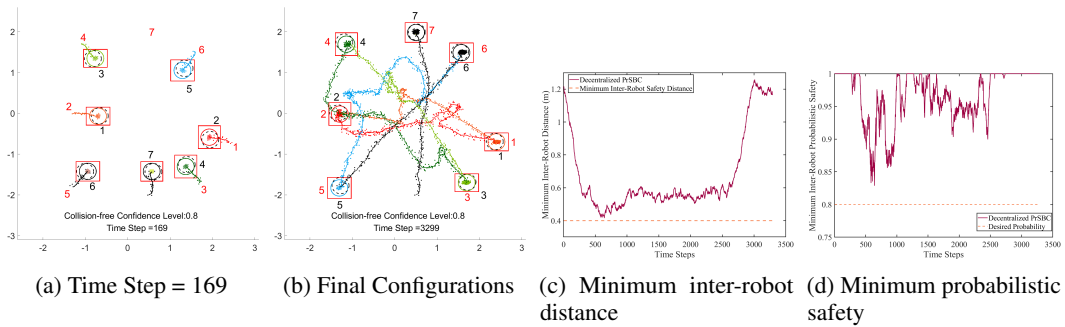

(a) Time Step = 169   (b) Final Configurations   (c) Minimum inter-robot distance   (d) Minimum probabilistic safety

Figure 2: Decentralized PrSBC with 7 robots. Robots 6 and 7 marked in black serve as passive moving obstacles without interaction to other robots.

motion disturbances. While with our PrSBC controller in (15), robots safely navigate through the work space (Fig. 1d) (but not too conservatively as it still allows interaction between bounding error box shown in Fig. 1b for probabilistic safety). In particular, results in Fig. 1h indicates our PrSBC method successfully ensures the satisfying probabilistic safety ($\sigma = 0.9$). This is computed by the minimum ratio between non-overlapping area and the area within robot's red bounding error box.

**Scenario with Dynamic Obstacles:** To account for dynamic obstacles, we add robot 7 to the previous scenario and make robot 6 and 7 serve as the non-cooperating passive moving obstacles. Fig. 2 highlights our observations from this experiment. We assume robots can identify them as obstacles instead of cooperating robots. With the same set-up except for the two obstacles, we demonstrate the performance of our controller based on decentralized PrSBC in (17) and set the inter-robot, robot-obstacle collision-free confidence $\sigma = \sigma_o = 0.8$ to encourage more flexible motion. In the decentralized settings, robots are set to assume equal responsibility in collision avoidance, i.e. $p_{ij} = p_{ji} = 0.5$ in (16) for each robot, and thus no communication is needed between robots. Results in Fig. 2c and 2d indicate the inter-robots and robot-obstacle are collision free and with a satisfying probabilistic safety close to $\sigma = 0.8$ (thus not overly conservative). From Fig. 2b it is noted that robot 5 with light blue trajectory took a large detour before reaching the goal position. This is caused by the non-cooperating obstacle robot 6 and 7 in the way, where the

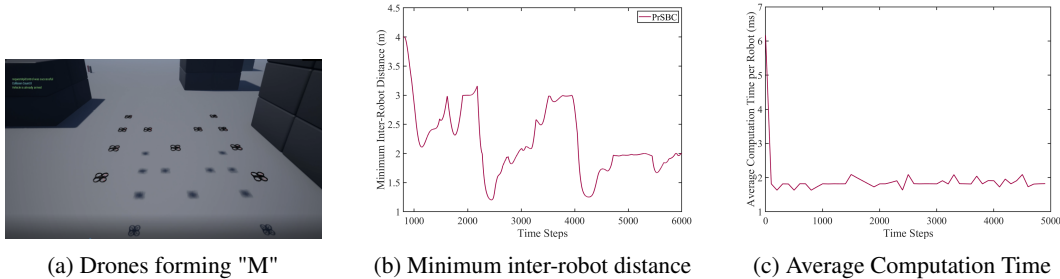

(a) Drones forming "M"  (b) Minimum inter-robot distance  (c) Average Computation Time

Figure 4: AirSim [21] experiment snapshot with 11 drones using our PrSBC for collision avoidance.

PrSBC for obstacles (14) forces the robot 5 to obey the more restrictive constraints to adapt to the momentum in order to guarantee the satisfying probabilistic collision avoidance performance.

**Quantitative Results:** We performed 50 random trials with different number of robots under a required confidence $\sigma = 0.9$ to validate the effectiveness of our decentralized PrSBC controller in presence of random measurement and motion noise. Fig. 3a and 3b shows that the robots are always safe and satisfy the probabilistic safety guarantee using PrSBC.

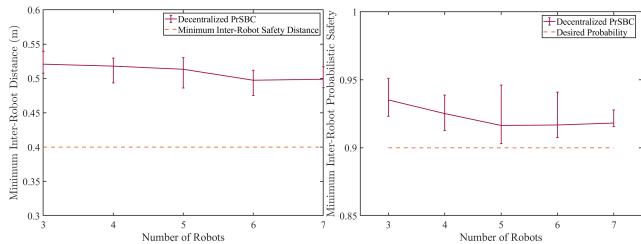

(a) Minimum inter-robot distance (b) Minimum probabilistic safety

Figure 3: Quantitative results summary of PrSBC from 50 random trials.

**Experimental Results:** Finally, as shown in Fig. 4, we carried out experiments with 11 simulated drones in AirSim [21], an open-source near-realistic simulation environment. The dynamics model of a quadrotor is a 12 dimensional under-actuated system [19, 29] and given its differential flatness property, vector-field based controller [32] could be employed to map the input velocity command to the quadrotor dynamics without compromising safety guarantee. The primary task for the drones is to sequentially form the letters of M-S-F-T while avoiding collisions with each other with the minimum probability of 0.9. Each of the drones has the pre-defined target position in the letter formation and they execute the gradient based controller to move towards it. The safety radius between pairwise drones is $1m$ and the state estimation noise is between $[-0.2m, 0.2m]$. We then employ our PrSBC controller to compute the linear velocity for each drone and feed it to the vector-based drone controller in the simulator. During the task, no collisions are observed as shown in Fig. 4b. The simulations are on personal laptop with Intel Core i7-8750H CPU of 2.20 GHz. The average computation time per robot is below $2ms$ as reported in Fig. 4c, demonstrating the efficiency of our PrSBC in real-time computation. Readers are encouraged to look to details of the experiments in the Video attachments.

## 7   Conclusions and Future Work

We presented a probabilistic approach to address chance constrained collision avoidance for a system of multiple robots in real-world settings. We address the complexities that arise due to uncertainty in perception and incompleteness in modeling the underlying dynamics of the system. The key idea is to induce probabilistic constraints via safety barriers, which are then used to minimally modify an existing controller via a constrained quadratic program. We formally define Probabilisitc Safety Barrier Certificates that guarantee forward-invariance in time continuously and also can be decomposed so as to enable de-centralized computation of the safe controllers. Future work entails extensions to model-free controllers trained via Reinforcement Learning and implementation to solve real-world tasks, such as Automatic Collision Avoidance System for manned and unmanned aircraft. We plan to employ variants of CBF such as ECBF [16] to explicitly handle higher order dynamics. On the other hand, extending the expressivity of the PrSBC formulation with different forms of the safe set $h^s(\mathbf{x})$ to address other uncertainty-aware safety consideration beyond collision avoidance is also an important future direction, e.g. limiting the number of drones within a volume, and adapting to temporal safety tasks using signal temporal logic (STL) formulations [15].

## Broader Impact

The objective of this work is to provide an explicit safety design for multi-robot systems in terms of collision avoidance that could guarantee probabilistic safety in real-world applications under uncertainty. This is a critical component towards AI and robotics safety [5] when we envision a future with significant increase on AI and multi-robot deployments to our society. As pointed out in [5], while we have seen successful efforts in the aircraft collision avoidance system, the same verification tools are often unable to be directly applied to modern autonomous system powered by AI and machine learning under uncertainty, e.g. autonomous drone fleets. And yet this technique is in high demands considering its wide applications that are rapidly growing at scale. The ultimate goal of this work is to develop such a model-based, formally provable automatic collision avoidance system (ACAS) for autonomous aerial robots that work with various uncertainty models developed by perception modules, AI and machine learning technologies, and to enable runtime verification and mitigation so that the executed control policies are safe at all times. An intuitive example is to consider the transfer of a control policy trained in a simulator to the real-world deployment. In this case, it is desired to have an unbiased *barrier* wrapped around the policy so that the safety is always ensured in the first place, which is the exact purpose of our proposed probabilistic safety barrier certificates (PrSBC) constraints. We believe our work will lead to fruitful results on safety improvements for both civil applications and academic research. On the other hand, as critical as the safety itself, the consequence of failure of such safety system could also be catastrophic in nature. For example, very inaccurate robot dynamics model and super inferior sensing information from the environments could cast immense threats to the safety design. We strive to minimize these factors by always accounting for uncertainty, properly leveraging conservativeness and absolute safety, and including worst-case analysis to increase the robustness of our design.

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
