[Supplementary Material]

# Multi-Robot Collision Avoidance under Uncertainty with Probabilistic Safety Barrier Certificates

## Appendix

*Equation indexes from (1)-(17) follow the original indexes appearing in the paper submission and new equations start from (18) in this document.

# A  Detailed Proofs

## A.1  Proof of Theorem 3

**Theorem 1.** ***Existence of PrSBC:*** *Assuming all pairwise robots are initially collision-free at $t = 0$, i.e. equ. (2) holds true for all possible value of random state variables $\mathbf{x}_i \in [\hat{\mathbf{x}}_i - \Delta\mathbf{v}_i, \hat{\mathbf{x}}_i + \Delta\mathbf{v}_i], \forall i \in \mathcal{I}$, then the PrSBC defined in equ. (10) is guaranteed to exist for any given confidence level $\sigma \in [0, 1]$.*

$$h_{i,j}^s(\mathbf{x}) = \|\mathbf{x}_i - \mathbf{x}_j\|^2 - (R_i + R_j)^2 \ , \ \forall i > j \tag{2}$$

$$\mathcal{S}_u^\sigma = \{\mathbf{u} \in \mathbb{R}^{mN} \mid A_{ij}^\sigma \mathbf{u} \le b_{ij}^\sigma, \quad \forall i > j, \ \mathbf{A}^\sigma \in \mathbb{R}^{n \times mN}, \ \mathbf{b}^\sigma \in \mathbb{R}^n \} \tag{10}$$

*Proof.* We start by proving the existence of PrSBC between each pairwise robots $i$ and $j$ with any user-defined confidence level $\sigma \in [0, 1]$. Given the sufficiency condition of $\mathrm{Pr}(\mathbf{u}_i, \mathbf{u}_j \in \mathcal{B}_{i,j}^s(\mathbf{x})) \ge \sigma$ in (9) with pairwise version of (8) rendering desired chance constrained safety $\mathrm{Pr}(\mathbf{x}_i, \mathbf{x}_j \in \mathcal{H}_{i,j}^s) \ge \sigma$:

$$\begin{aligned}\mathcal{B}^s(\mathbf{x}) &= \{\mathbf{u} \in \mathbb{R}^{mN} : \dot{h}_{i,j}^s(\mathbf{x}, \mathbf{u}) + \gamma h_{i,j}^s(\mathbf{x}) \ge 0, \ \forall i > j\} \\ \mathcal{B}_{i,j}^s(\mathbf{x}) &= \{\mathbf{u}_i, \mathbf{u}_j \in \mathbb{R}^m : \dot{h}_{i,j}^s(\mathbf{x}, \mathbf{u}) + \gamma h_{i,j}^s(\mathbf{x}) \ge 0\}\end{aligned} \tag{8}$$

$$\mathrm{Pr}(\mathbf{u}_i, \mathbf{u}_j \in \mathcal{B}_{i,j}^s(\mathbf{x})) \ge \sigma \implies \mathrm{Pr}(\mathbf{x}_i, \mathbf{x}_j \in \mathcal{H}_{i,j}^s) \ge \sigma, \quad \forall i > j \tag{9}$$

Consider $\dot{h}_{i,j}^s(\mathbf{x}, \mathbf{u}) = \frac{\partial h_{i,j}^s}{\partial \mathbf{x}}(\mathbf{x})(\Delta F_{i,j}(\mathbf{x}) + G_{i,j}(\mathbf{x})\mathbf{u}_{i,j} + \Delta\mathbf{w}_{i,j})$ , we can then re-write the sufficiency condition $\mathrm{Pr}(\mathbf{u}_i, \mathbf{u}_j \in \mathcal{B}_{i,j}^s(\mathbf{x})) \ge \sigma$ in (9) using (8) as follows:

$$\begin{aligned}\mathrm{Pr}(\mathbf{u}_i, &\mathbf{u}_j \in \mathcal{B}_{i,j}^s(\mathbf{x})) \ge \sigma : \\ &\iff \mathrm{Pr}\left(\frac{\partial h_{i,j}^s}{\partial \mathbf{x}}(\mathbf{x})G_{i,j}(\mathbf{x})\mathbf{u}_{i,j} \ge -\gamma h_{i,j}^s(\mathbf{x}) - \frac{\partial h_{i,j}^s}{\partial \mathbf{x}}(\mathbf{x})\big(\Delta F_{i,j}(\mathbf{x}) + \Delta\mathbf{w}_{i,j}\big)\right) \ge \sigma\end{aligned} \tag{11}$$

where

$$\begin{aligned}\frac{\partial h_{i,j}^s}{\partial \mathbf{x}}(\mathbf{x})G_{i,j}(\mathbf{x})\mathbf{u}_{i,j} &= 2(\mathbf{x}_i - \mathbf{x}_j)^T\left(G_i(\mathbf{x}_i)\mathbf{u}_i - G_j(\mathbf{x}_j)\mathbf{u}_j\right) \\ \frac{\partial h_{i,j}^s}{\partial \mathbf{x}}(\mathbf{x})\left(\Delta F_{i,j}(\mathbf{x}) + \Delta\mathbf{w}_{i,j}\right) &= 2(\mathbf{x}_i - \mathbf{x}_j)^T\left(F_i(\mathbf{x}_i) - F_j(\mathbf{x}_j) + \mathbf{w}_i - \mathbf{w}_j\right)\end{aligned} \tag{18}$$

Let's denote the process noise difference $\Delta\mathbf{w}_{i,j} = \mathbf{w}_i - \mathbf{w}_j \sim Q_{i,j}$ with the finite support $\mathrm{supp}(Q_{i,j}) = \left[-(\Delta\mathbf{w}_i + \Delta\mathbf{w}_j), \ (\Delta\mathbf{w}_i + \Delta\mathbf{w}_j)\right]$ and state difference $\Delta\mathbf{x}_{i,j} = \mathbf{x}_i - \mathbf{x}_j \sim T_{i,j}$ with the finite support $\mathrm{supp}(T_{i,j}) = \left[(\hat{\mathbf{x}}_i - \hat{\mathbf{x}}_j) - (\Delta\mathbf{v}_i + \Delta\mathbf{v}_j), \ (\hat{\mathbf{x}}_i - \hat{\mathbf{x}}_j) + (\Delta\mathbf{v}_i + \Delta\mathbf{v}_j)\right]$. Moreover, given the assumed uniform distributions of $\mathbf{w}_i, \mathbf{w}_j, \mathbf{x}_i, \mathbf{x}_j$, the distributions $T_{i,j}, Q_{i,j}$ are hence two different symmetric trapezoid

distributions with finite supports. Then by substituting (18) into (11) and after re-organization, we have

$$\Pr\left(\left[\Delta\mathbf{x}_{i,j}+\overbrace{\frac{G_{i,j}\mathbf{u}_{i,j}+\Delta F_{i,j}+\Delta\mathbf{w}_{i,j}}{\gamma}}^{\dot{\mathbf{x}}_i-\dot{\mathbf{x}}_j}\right]^2 \geq R_{ij}^2+\left[\overbrace{\frac{G_{i,j}\mathbf{u}_{i,j}+\Delta F_{i,j}+\Delta\mathbf{w}_{i,j}}{\gamma}}^{\dot{\mathbf{x}}_i-\dot{\mathbf{x}}_j}\right]^2\right) \geq \sigma \qquad (19)$$

where

$$G_{i,j}\mathbf{u}_{i,j}=G_i(\mathbf{x}_i)\mathbf{u}_i - G_j(\mathbf{x}_j)\mathbf{u}_j\,,\quad \Delta F_{i,j}=F_i(\mathbf{x}_i)-F_j(\mathbf{x}_j)\,,\quad R_{ij}=R_i+R_j>0 \qquad (20)$$

Thus consider the following set of random variable $\Delta\mathbf{x}_{i,j}$ from its own finite support and (19):

$$\Omega_{i,j}(\Delta\mathbf{x}_{i,j})=\mathrm{supp}(T_{i,j})=\left[(\hat{\mathbf{x}}_i-\hat{\mathbf{x}}_j)-(\Delta\mathbf{v}_i+\Delta\mathbf{v}_j),\quad (\hat{\mathbf{x}}_i-\hat{\mathbf{x}}_j)+(\Delta\mathbf{v}_i+\Delta\mathbf{v}_j)\right]$$

$$\Omega_{i,j}^{\mathbf{u}}(\Delta\mathbf{x}_{i,j})=\left\{\Delta\mathbf{x}_{i,j}\in\mathbb{R}^d\;\middle|\;\left[\Delta\mathbf{x}_{i,j}+\overbrace{\frac{G_{i,j}\mathbf{u}_{i,j}+\Delta F_{i,j}+\Delta\mathbf{w}_{i,j}}{\gamma}}^{\dot{\mathbf{x}}_i-\dot{\mathbf{x}}_j}\right]^2 \geq R_{ij}^2+\left[\overbrace{\frac{G_{i,j}\mathbf{u}_{i,j}+\Delta F_{i,j}+\Delta\mathbf{w}_{i,j}}{\gamma}}^{\dot{\mathbf{x}}_i-\dot{\mathbf{x}}_j}\right]^2\right\} \qquad (21)$$

Note that the set of $\Omega_{i,j}^{\mathbf{u}}(\Delta\mathbf{x}_{i,j})$ representing the space outside a $(d-1)-$sphere for $\Delta\mathbf{x}_{i,j}$ in $d-$dimensional space. It is determined by the pairwise value of $\mathbf{u}_i,\mathbf{u}_j$ through $G_{i,j}\mathbf{u}_{i,j}=G_i(\mathbf{x}_i)\mathbf{u}_i - G_j(\mathbf{x}_j)\mathbf{u}_j$ as defined in (20). It is thus straightforward to show that the condition in (19) is equivalent to:

$$\Pr\left(\Delta\mathbf{x}_{i,j}\in\Omega_{i,j}\cap\Omega_{i,j}^{\mathbf{u}}\right)\geq\sigma \qquad (22)$$

To prove the guaranteed existence of PrSBC, we need to show there always exists at least one solution of pairwise $\mathbf{u}_i,\mathbf{u}_j$ such that (22) holds for any given value of $\sigma\in[0,1]$. First let's consider any pairwise $\mathbf{u}_i=\mathbf{u}_i^0,\mathbf{u}_j=\mathbf{u}_j^0$ leading to the joint control inputs $\mathbf{u}^0$ such that $\dot{\mathbf{x}}_i-\dot{\mathbf{x}}_j=0$ in (21), then we have the following condition representing the space outside the $(d-1)-$sphere for $\Delta\mathbf{x}_{i,j}$:

$$\Omega_{i,j}^{\mathbf{u}^0}(\Delta\mathbf{x}_{i,j})=\left\{\Delta\mathbf{x}_{i,j}\in\mathbb{R}^d\;\middle|\;\Delta\mathbf{x}_{i,j}^2\geq R_{ij}^2\right\} \qquad (23)$$

Recall that all pairwise robots are assumed to be initially collision-free, i.e. $\Delta\mathbf{x}_{i,j}^2\geq R_{ij}^2$, thus (22) holds true at all times for any given $\sigma\in[0,1]$ since $\Pr\left(\Delta\mathbf{x}_{i,j}\in\Omega_{i,j}\cap\Omega_{i,j}^{\mathbf{u}^0}\right)=1$ under one possible solution of joint control inputs $\mathbf{u}=\mathbf{u}^0$ that leads to $\dot{\mathbf{x}}_i-\dot{\mathbf{x}}_j=0$. More generally, as the value of $\|\dot{\mathbf{x}}_i-\dot{\mathbf{x}}_j\|$ grows from 0 with other value of $\mathbf{u}\neq\mathbf{u}^0$, the corresponding $(d-1)-$sphere of $\Omega_{i,j}^{\mathbf{u}}(\Delta\mathbf{x}_{i,j})$ in (21) will continuously shift from the origin and gradually intersect with the bounding box of $\Omega_{i,j}(\Delta\mathbf{x}_{i,j})=\mathrm{supp}(T_{i,j})$ in (21). This leads to $\Pr\left(\Delta\mathbf{x}_{i,j}\in\Omega_{i,j}\cap\Omega_{i,j}^{\mathbf{u}}\right)$ continuously decrease from 1 to 0. Hence for any given value $\sigma\in[0,1]$, it is always feasible to solve for at least a particular pairwise $\mathbf{u}_i,\mathbf{u}_j$ such that $\Pr\left(\Delta\mathbf{x}_{i,j}\in\Omega_{i,j}\cap\Omega_{i,j}^{\mathbf{u}}\right)=\sigma$, or $\Pr\left(\Delta\mathbf{x}_{i,j}\in\Omega_{i,j}\cap\Omega_{i,j}^{\mathbf{u}}\right)>\sigma$ so that (22) holds true. This pairwise $\mathbf{u}_i,\mathbf{u}_j$ could then serve as a hyperplane dividing the corresponding subspace of joint control space of $\mathbf{u}$ with one side in the form of (10) rendering the satisfying probabilistic safety between robot $i,j$. And by repeatedly updating the hyperplane at each time step in (10), the constrained step-wise controllers $\mathbf{u}_i,\mathbf{u}_j$ ensure the probabilistic safety is guaranteed at all times given the forward invariance in (9). It is then straightforward to extend to all pairwise inter-robot collision avoidance constraints and thus concludes the proof. □

## A.2  Computation of PrSBC

In this part we will provide computation of PrSBC that yield the solution in equation (12) and (13). Lets consider inter-robot collision avoidance first. Given any confidence level $\sigma \in [0, 1]$, from Section A.1 the equivalent chance constraint of $\Pr(\mathbf{u}_i, \mathbf{u}_j \in \mathcal{B}_{i,j}^s(\mathbf{x})) \geq \sigma$ in (11) and its re-written form in (19) can be transformed into a deterministic linear constraint over pairwise controllers $\mathbf{u}_i, \mathbf{u}_j$ in the form of (10). While it is computationally intractable to get closed form solutions from (19), we obtain an approximate solution by considering the condition on each individual dimension $\Delta \mathbf{x}_{i,j}^l \in \{\Delta \mathbf{x}_{i,j}^1, \ldots, \Delta \mathbf{x}_{i,j}^d\} \subset \mathbb{R}^d$ of $\Delta \mathbf{x}_{i,j}, \forall l = 1, \ldots, d$ for (19). Hence, we introduce a sufficiency condition to (19) in each dimension as follows, so that ensuring (24) $\Longrightarrow$ (19).

$$\Pr\left( (\Delta \mathbf{x}_{i,j}^l)^2 + 2 \cdot \frac{(G_{i,j}\mathbf{u}_{i,j})_l + \Delta F_{i,j}^l}{\gamma} \Delta \mathbf{x}_{i,j}^l \geq R_{ij}^2 - B_{i,j}^l \right) \geq \sigma \tag{24}$$

where $(G_{i,j}\mathbf{u}_{i,j})_l = (G_i\mathbf{u}_i - G_j\mathbf{u}_j)_l \in \mathbb{R}$ and $\Delta F_{i,j}^l = F_i^l - F_j^l \in \mathbb{R}$ denote the $l$th element of $G_{i,j}\mathbf{u}_{i,j} \in \mathbb{R}^{d \times 1}$ and $\Delta F_{i,j} \in \mathbb{R}^{d \times 1}$ respectively. $B_{i,j}^l = -\frac{2}{\gamma}\max \|\Delta \mathbf{w}_{i,j}^l\| \cdot \|\Delta \mathbf{x}_{i,j}^l\| \in \mathbb{R}$ with $\Delta \mathbf{w}_{i,j}^l \in \mathbb{R}$ as the $l$th element in $\Delta \mathbf{w}_{i,j} \in \mathbb{R}^d$. To simplify the discussion we assume piece-wise $G_i, G_j \in \mathbb{R}^{d \times m}, F_i, F_j \in \mathbb{R}^{d \times 1}$ in (1) are known and deterministic. Then, we have equivalent condition of (24) as follows

$$\Pr\left( \Delta \mathbf{x}_{i,j}^l \leq -\frac{(G_{i,j}\mathbf{u}_{i,j})_l + \Delta F_{i,j}^l}{\gamma} - D_{i,j}^l \ \text{ OR } \ \Delta \mathbf{x}_{i,j}^l \geq -\frac{(G_{i,j}\mathbf{u}_{i,j})_l + \Delta F_{i,j}^l}{\gamma} + D_{i,j}^l \right) \geq \sigma \tag{25}$$

where

$$D_{i,j}^l = \sqrt{\frac{\left((G_{i,j}\mathbf{u}_{i,j})_l + \Delta F_{i,j}^l\right)^2}{\gamma^2} + R_{ij}^2 - B_{i,j}^l}$$

Recall the finite support of $\Delta \mathbf{x}_{i,j}$ with its symmetric trapezoid distribution $T_{i,j}$ in (21), we can find alternative condition to enforce either of the condition in (25), e.g. $\Pr(\Delta \mathbf{x}_{i,j}^l \leq \cdot) \geq \sigma$ or $\Pr(\Delta \mathbf{x}_{i,j}^l \geq \cdot) \geq \sigma$ so that (25) is definitely lower bounded by $\sigma$. We assume $\sigma > 0.5$ and denote $e_{i,j}^{l,1} = \Phi^{-1}(\sigma)$ and $e_{i,j}^{l,2} = \Phi^{-1}(1 - \sigma)$ with $\Phi^{-1}(\cdot)$ as the inverse cumulative distribution function (CDF) of the random variable $\Delta \mathbf{x}_{i,j}^l = \mathbf{x}_i^l - \mathbf{x}_j^l$ in (18) along each $l$th dimension. We have $\sigma > 0.5 \implies e_{i,j}^{l,1} > e_{i,j}^{l,2}$. Thus, we derive a formal sufficiency condition for (25) as follows.

$$\exists l = 1, \ldots, d: \quad -2\mathbf{e}_{i,j}^l(G_{i,j}\mathbf{u}_{i,j})_l/\gamma \leq (\mathbf{e}_{i,j}^l)^2 - R_{ij}^2 + B_{ij}^l + 2\mathbf{e}_{i,j}^l \Delta F_{i,j}^l/\gamma \tag{12}$$

where

$$\mathbf{e}_{i,j}^l = \begin{cases} e_{i,j}^{l,2}, & e_{i,j}^{l,2} > 0 \\ e_{i,j}^{l,1}, & e_{i,j}^{l,1} < 0 \\ 0, & e_{i,j}^{l,2} \leq 0 \text{ and } e_{i,j}^{l,1} \geq 0 \end{cases}$$

Note that $\mathbf{e}_{i,j}^l = 0$ implies the two robots $i$ and $j$ overlap along the $l$th dimension, e.g. two drones flying to the same 2D locations but with different altitudes. As it is assumed any pairwise robots are initially collision free and from the forward invariance property discussed above, $\mathbf{e}_{i,j}^l = 0$ only happens along at most $d - 1$ dimensions. To that end, we can formally construct the PrSBC as in (10) with the following linear deterministic constraints in closed form.

$$\mathcal{S}_u^\sigma = \{\mathbf{u} \in \mathbb{R}^{mN} | -2\mathbf{e}_{i,j}^T(G_i\mathbf{u}_i - G_j\mathbf{u}_j)/\gamma \leq \|\mathbf{e}_{i,j}\|^2 - d \cdot R_{ij}^2 + B_{ij} + 2\mathbf{e}_{i,j}^T \Delta F_{i,j}/\gamma, \quad \forall i > j\} \tag{13}$$

where $\mathbf{e}_{i,j} = [\mathbf{e}_{i,j}^1, \ldots, \mathbf{e}_{i,j}^d]^T \in \mathbb{R}^{d \times 1}$ and $B_{ij} = \Sigma_{l=1}^d B_{ij}^l$. This invokes a set of pairwise linear constraints over the robot controllers such that the inter-robot probabilistic collision avoidance in (4) holds true at all times. Note the PrSBC constraint in (13) is a conservative approximation of (12) by adding up the constraints for each dimension, and therefore guarantee $\Pr(\mathbf{u}_i, \mathbf{u}_j \in \mathcal{B}_{i,j}^s(\mathbf{x})) \geq \sigma$.