[Reviews · NeurIPS 2020]

Review 1

Summary and Contributions: This paper proposes the extension of safety barrier certificates to consider uncertainty in multi-robot systems, in particular they provide a system that satisfies collision-avoidance chance constraints.

Strengths: The topic of the paper is quite relevant as safety barrier certificates have attracted quite a lot of attention from the control community and extending them the inherent uncertainty of robot systems is a worthwhile endeavour. Furthermore the formalisation looks elegant to me, even though I’m not an expert in the field of control.

Weaknesses: The chance constraints are imposed on a per timestep basis, but there’s no notion of probability of collision for a full trajectory of the robots, nor is that discussed. I’d assume that with chance constraints lower than 1 one will be building systems that are surely brittle in the long run, as the probability of collision will tend to 1 as the number of conflicts between robots goes to infinity. Furthermore, the experimental section doesn't explicitly consider different levels of conservativeness, which could allow the reader to have some extra insight on the former point. The paper is quite relevant to the control community, but I have some doubts on its relevance to the NeurIPS community. In particular, the proposed approach has no learning component at all and is very much pure control. I’d expect the control theory papers in NeurIPS to tackle problems like partially known dynamics for example, and this is nothing of that sort.

Correctness: As far as I understand, the paper is correct.

Clarity: Yes, the paper is well organised and written.

Relation to Prior Work: As far as I am concerned the related work section looks good, but I’m completely unaware of the research on safety barrier certificates.

Reproducibility: Yes

Additional Feedback: Equations 2 and 3 are pretty much the same, so it’s a bit redundant writing both down. After the rebuttal phase, I'm happy with the general response the authors gave regarding relevance to NeurIPS. However, given that most in the NeurIPS community will not be familiar with SBCs, I think that 1) the discussion given regarding integration with learnt models should be provided in the paper itself; and 2) more care should be given to the presentation of SBCs to ensure the paper is sufficiently self contained and there are no assumptions on prior knowledge on the topic. Given this, I keep my opinion that this is a very borderline paper.


Review 2

Summary and Contributions: This paper proposed PrSBC, which extended the safety barrier certification in probabilistic settings, and used it to guarantee safety (collision avoidance) for robotic applications. This method formulates the safety controller as a constrained QP, which minimally changes the nominal control input, and subject to linear constraints derived from PrSBC. The paper compared PrSBC with SBC in two simple simulated environments and showed no safety violations when the noises are present.

Strengths: The paper has the two following strengths: 1) Since the real world is inherently stochastic, it is extremely important to reason about safety by taking into consideration noise and uncertainty. For this reason, extending a popular safety algorithm (SBC) to probabilistic settings is a significant contribution. 2) The proposed approach can be generalized to different forms of uncertainty. In theory, its derivation can work with any noise models with finite support.

Weaknesses: In my personal opinion, the weaknesses of this paper are: 1) The algorithm is evaluated in two relatively simple scenarios. It is not clear to me the generality of the method. For example, can the proposed algorithm be applied to safety control problems in a higher dimensional space (e.g. self collision avoidance for a 7-dof robot arm) or with more realistic dynamics (e.g. heavy unicycle with low ground friction). 2) Parts of the paper is not crystal clear to me. Please see "Clarity" for more details.

Correctness: The high-level derivations of the method seem reasonable. I did not carefully check the details or the proofs in the Appendix, though.

Clarity: The paper is mostly well written. The accompanying video is helpful to visualize the results. I have one major confusion though, which probably is due to my ignorance of the background of Safety Barrier Certificates. Since many readers of NeurIPS papers may also not be experts in this field, it would be great to give more detailed explanations that may eliminate this potential confusion: I believe that the optimization (15) is to find a safe u for the current time step. This optimal u is not the entire trajectory over time. In the next time step, the QP is formed and solved again. If this is the case, how long are the PrSBC constraints valid? In other words, if u belongs to S_u^\sigma, does it guarantee that collision will not happen in the next time step, or for the next few time steps, or forever? The discussion about "forward-invariant" in Section 4 suggests that the certificate is a subspace of all possible controls in the current time step that can guarantee no collision forever in the future (all t > 0). Does it mean that if I choose u \in S_u^\sigma, collision cannot happen in the future? This would be too good to be true. Otherwise, does it mean that if I choose u \in S_u^\sigma, collision cannot happen in the near future (e.g. next 10 time steps)? If so, what is the time horizon that safety is guaranteed? Which parameter controls this time horizon?

Relation to Prior Work: The relation to prior wok is clearly and sufficiently discussed.

Reproducibility: Yes

Additional Feedback: ----------------Post rebuttal comments----------------- Thanks for the response. It addressed my questions.


Review 3

Summary and Contributions: The main contribution of this paper is to apply the Barrier Certificate formulation of safety constraint to the problem of multi-robot collision-free path planning via optimization methods. This is demonstrated using the example of multiple UAVs performing coordinated motion planning in a simulation environment.

Strengths: The main feature of this paper is the application of the barrier certificates methodology in the context of multiple UAV path planning, taking into account uncertainty in perception and actions. So, the authors adopt a probabilistic version of barrier certificates, defining it in terms of level sets of confidence. So, the authors derive a chance constraint over pairs of controllers, in order to enforce minimum distance limits etc. Given a desired level of probability, this is then turned into a linear constraint in space which can then be enforced in control computation. This then facilitates optimization based trajectory generation, which is a contribution, and also decentralized schemes based on defining a common protocol for how different robots interact with each other.

Weaknesses: While the methodology is overall quite sound, I am unsure about two main points: (1) The overall approach is to gradually reduce the various forms of variability in the problem. So, for instance, we start with a chance constraint formulation which point-wise and pair-wise delineates the concerns of collision-avoidance. Then, we have a specific protocol for turning the problem into a decentralised form, etc. All this limits the expressivity of the overall framework. So, for instance, if the overall constraint were not just collision avoidance - say, we had a max restriction on numbers of agents allowed within a volume (timely, as I write this review!) - then it is not clear there is an easy adaptation. So, in this sense, the paper seems quite closely tied to the specific case of collision-avoidance rather than more general forms of safety. To give a different example, how would we compile the constraints similarly if the safety constraint were a temporal statement such as needing to visit a charging station at some point within every 5 min window? I would have appreciated a more detailed discussion of such issues around modelling. (2) The literature on distributed control of robots (e.g., http://coordinationbook.info/) includes several methods for achieving spatial configurations that also admit dynamical systems analysis for convergence and correctness akin to the formalism here. Those authors would not have phrased their claims in the same way but I am sure the present authors can see that the bounds being used to define SBCs here are also the same as those known in dynamical systems. Indeed, even some of these concerns about decentralisation have been considered before, and the tasks shown in fig 2 are familiar in that setting. If so, how do the methods compare and to the sceptic how much of what is shown here is new? I am not necessarily claiming there isn't novelty but the paper would be stronger by clearly discussing the comparisons.

Correctness: The arguments in this paper seem to me to be correct.

Clarity: The paper is well written. I have offered comments above to suggest improvements, but I was able to follow the arguments reasonably well.

Relation to Prior Work: I believe the authors have appropriately cited related work. I have made a few suggestions above for the authors to consider.

Reproducibility: Yes

Additional Feedback: [Post-author-response comments] Thanks for addressing some of my questions, and I hope this will be expanded upon within the main text of the paper.


Review 4

Summary and Contributions: The paper contributes by presenting an approach for the collision avoidance problem in multi-robot systems, which considers the uncertainty in sensors’ measurements and the inaccuracy of the robots’ model to provide guarantees on safety when computing motion controls for the robots. The problem is well-defined and formally presented. The Related Work section shows relevant and recent works on collision avoidance for multi-robot systems. Safety is defined as a function of distance between pairs of robots. Then, the probability that the robots maintain this safety distance is defined to be above a certain level σ (this is a parameter given by the end user). By following the control barrier functions formulation, in a previous work the authors presented the Safety Barrier Certificates, defined as the set of admissible controls for the robots, computed at each time step, which are collision-free at all times. From this formulation, the authors present the Probabilistic Safety Barrier Certificates, to compute the set of admissible controls that ensure the probability of the robots to avoid collision is above or equal to σ. This formulation translates into defining, for each robot, the intersection of half-spaces between pairs of robots and robot-obstacles in the joint control space. The approach is theoretically proven. From this resultant set of admissible controls, an optimization problem is solved to compute, for each robot, the nearest control from a nominal control (it is assumed the robots are moving towards their goal by following a task-related control). Both a centralized and decentralized versions are presented. In the latter, the effort that each robot makes to avoid the collision has to be considered, which translates into restricting the set of admissible controls. The evaluation of the system demonstrates the suitability of the approach.

Strengths: Theoretical validation of the approach. Thorough evaluation through simulation of the solution proposed.

Weaknesses: The authors could elaborate more on the probabilistic model used for evaluation.

Correctness: Correctness is observed in the claims, methodology and evaluation.

Clarity: The paper is very well written, and easy to understand. Though there are few grammatical errors to amend.

Relation to Prior Work: The paper extends on previous work, though the authors clearly differentiate the contribution with respect to the previous work.

Reproducibility: Yes

Additional Feedback: The rebuttal has been taking into account. We are happy in general with the information that was provided. We agree with other reviewers that a more extensive simulation would further strengthen the work. Minor issues to be amended: - Sometimes the space of control is indicated as R^d (in the main paper) and R^m (in the appendix and paper), which is the correct one? - Missing “/” in formula 16, the second part: “eij / γ · uj ≤ pji(pij + pji) · bσij ”, should be “eij / γ · uj ≤ pji / (pij + pji) · bσij ” - In line 248, “∀ j > i” should be “∀ i > j”, for consistency with the rest of formulation in the paper. - In the video, in the example “2 unicycle robots swap positions”, the size of the box for robot 1 is bigger than the box for robot 2. Is this because the error is greater? How you select the error distribution for each of the robots?

[Author Response · NeurIPS 2020]

We thank reviewers for the constructive comments. Reviewers found our paper "(very) well written" (R1-R4), "a
significant contribution" (R2), "a worthwhile endeavour" (R1), and "quite sound" (R3). Please find our response below.

**[R1] Probability of collision for a full trajectory**: Given $n_t$ as the total number of time steps, the probability of
collision avoidance between robot $i, j$ for the whole trajectory is lower bounded as $\Pr\big(\bigcap_{t=1}^{n_t}(\mathbf{x}_i^t, \mathbf{x}_j^t \in \mathcal{H}_{i,j}^s(t))\big) =$
$\prod_{t=1}^{n_t}\Pr(\mathbf{x}_i^t, \mathbf{x}_j^t \in \mathcal{H}_{i,j}^s(t)) \geq \sigma^{n_t}$. In theory, by selecting $\sigma = \exp(\frac{\ln\sigma_{all}}{n_t})$ one could achieve a lower bounded joint
collision free threshold of $\sigma_{all}$ for the full trajectory. However, it could be over conservative in the long run, e.g.
step-wise threshold $\sigma = 0.9949$ leads to $\sigma_{all} = 0.6$ for $n_t = 100$. Hence step-wise threshold is used more often to
construct local collision constraints (see [8,9,13,29,30,35,36]). An alternative is to impose discounting factor $\beta < 1$ so
that the penalty of future violation probabilities is relaxed, i.e. step-wise threshold $\sigma$ renders the same bounded joint
threshold for the whole trajectory $\sum_{t=1}^{n_t}(\beta)^t\Pr(\mathbf{x}_i^t, \mathbf{x}_j^t \in \mathcal{H}_{i,j}^s(t)) \geq \sigma$ if given discounting factor $\beta > 0.5$ (see [36]).
We will provide new results in the updated version with different conservativeness levels of $\sigma, \sigma_o$ to give more insights.

**[R1] Relevance to NeurIPS community**: Besides control community, we believe the work is well connected to
NeurIPS community as well. As the real world is inherently stochastic, it is important to derive mathematically correct
safety consideration accounting for uncertainties. In particular, we think the presented work could bridge learning based
methodologies and model based safety-critical control with formally provable safety guarantee. For example, a very
recent work [37] presents an episodic learning framework to model the partially known dynamics and uses control
barrier functions similar to the *deterministic* version of our approach for safety consideration *without* noisy uncertainties.
Another example: one may use learning techniques such as Gaussian Processes to learn a partially unknown dynamical
system with noisy uncertainties and use our approach to compute a certified probablistically safe policy to collect more
data for further improving models. We think integrating dynamical system learning with our PrSBC framework to
guarantee safe learning to control is an important future direction.

**[R2] Generality of the method**: Our model is general and can be applied to stochastic dynamical systems in control
affine form (1) which captures a large family of dynamical systems, such as 3-dof unicycle dynamics [19,28], 12-dof
quadrotors [29,33] (both already evaluated in our experiments), bipedal robots, automotive vehicle, and Segway robots
[4,37]. For heavy unicycle with low ground friction, our method is applicable as the robot could be described by a
nominal unicycle dynamics with limited acceleration [28] due to inertia and low friction with noises.

**[R2] Step-wise QP optimization**: The reviewer is correct about our step-wise optimization process (15). Our safe
controller $\mathbf{u}(\mathbf{x})$ is dependent on each visited robot state $\mathbf{x}$ and if the safe control $\mathbf{u}(\mathbf{x}(t)) \in \mathcal{S}_{\mathbf{u}}^\sigma \bigcap \mathcal{S}_{\mathbf{u}}^{\sigma_o}$ for all $t \in [0, \tau]$,
then it has chance constrained guarantee for any $\mathbf{x}(t) \in \mathcal{H}^s$ within $t \in [0, \tau]$, which folllows the definition of the
forward-invariance (see our Lemma 1 and Theorem 2 in [4]). From a continuous-time perspective, QP with PrSBC
constraints in (15) per time step ensures for all $t \in [0, \tau]$, $\mathbf{u}(\mathbf{x}(t)) \in \mathcal{S}_{\mathbf{u}}^\sigma \bigcap \mathcal{S}_{\mathbf{u}}^{\sigma_o}$, then our approach guarantees chance
constrained safety along the entire time horizon $[0, \tau]$, not just a particular time point.

**[R3] Expressivity of the framework**: While we address safety on collision avoidance by enforcing minimum distance
to define the control barrier function $h^s$ (2), the existence of PrSBC constraints in our Theorem 3 are general and do
not rely on the form of $h$ (see proof and (11)). For other safety consideration, we can apply our algorithm to different
forms of task-specific control barrier function $h$ as in Section V in [4] (see V.D for battery charging constraint example).
Recent work [38] has employed the deterministic control barrier functions to signal temporal logic (STL) formulations.
Our approach could be used in the same way with STL for explicit temporal safety considerations with uncertainty.

**[R3] Comparison to distributed robot control**: As discussed in [4], the traditional Lyapunov approach (as in the
referred book) handles stability that drives a dynamical system to a point or a overly restrictive sublevel set describing
*final* spatial configurations. However, safety is often framed as *enforcing invariance* of a permissive set, i.e. starting
from and not leaving a safe set. Furthermore, our contribution is the novel PrSBC framework that extends deterministic
safety barrier certificate approach to a probabilistic setting with formally proved bounded safety guarantee. This is
significantly different from the work in the book that does not address uncertainties, safety, nor collision avoidance.

**[R4] Probabilistic model used for evaluation**: In evaluation we use uniform distributions with finite support as the
probabilistic model and randomly sample from the distributions per time step to simulate the stochastic dynamics and
observations. The control space is $\mathbb{R}^m$ with $m \leq d$. In the example 2, the larger error box for robot 1 indicates a greater
uniform error with larger bounded support, which is randomly selected for testing purposes.

[37] Taylor, Andrew, Andrew Singletary, Yisong Yue, and Aaron Ames. "Learning for safety-critical control with control barrier
functions." In *Learning for Dynamics and Control*, pp. 708-717. 2020.

[38] Lindemann, Lars, and Dimos V. Dimarogonas. "Control barrier functions for signal temporal logic tasks." *IEEE control systems*
*letters* 3.1 (2018): 96-101.


[Meta-Review · NeurIPS 2020]

Beyond the strong support from R4, all reviewers recommend acceptance. Still, the reviewers raise some concerns that the authors should clarify for future versions of the paper, such as R1's concerns about the per-timestep basis of the high-probability guarantees allowing for high chances of failure in the long term. A reviewer also raises the question of whether NeurIPS is an appropriate venue for this work. Given the emphasis on safe reinforcement learning and the many methods in that field that try to tackle similar problems, this paper seems relevant and of interest to the NeurIPS community despite the control theory / non-learning nature of the proposed solution.